# Human-Based New Approach Methodologies in Developmental Toxicity Testing: A Step Ahead from the State of the Art with a Feto–Placental Organ-on-Chip Platform

**DOI:** 10.3390/ijerph192315828

**Published:** 2022-11-28

**Authors:** Michaela Luconi, Miguel A. Sogorb, Udo R. Markert, Emilio Benfenati, Tobias May, Susanne Wolbank, Alessandra Roncaglioni, Astrid Schmidt, Marco Straccia, Sabrina Tait

**Affiliations:** 1Department of Experimental and Clinical Biomedical Sciences “Mario Serio”, University of Florence, Viale Pieraccini 6, 50139 Florence, Italy; 2I.N.B.B. (Istituto Nazionale Biostrutture e Biosistemi), Viale Medaglie d’Oro 305, 00136 Rome, Italy; 3Instituto de Bioingeniería, Universidad Miguel Hernández de Elche, Avenida de la Universidad s/n, 03202 Elche, Spain; 4Placenta Lab, Department of Obstetrics, University Hospital Jena, Am Klinikum 1, 07747 Jena, Germany; 5Department of Environmental Health Sciences, Istituto di Ricerche Farmacologiche Mario Negri IRCCS, Via Mario Negri 2, 20156 Milan, Italy; 6InSCREENeX GmbH, Inhoffenstr. 7, 38124 Braunschweig, Germany; 7Ludwig Boltzmann Institut for Traumatology, The Research Center in Cooperation with AUVA, Austrian Cluster for Tissue Regeneration, Donaueschingenstrasse 13, 1200 Vienna, Austria; 8FRESCI by Science&Strategy SL, C/Roure Monjo 33, Vacarisses, 08233 Barcelona, Spain; 9Centre for Gender-Specific Medicine, Istituto Superiore di Sanità, 00161 Rome, Italy

**Keywords:** new approach methodologies (NAMs), adverse outcome pathways (AOPs), endocrine disruptors, animal replacement, OECD TG 414

## Abstract

Developmental toxicity testing urgently requires the implementation of human-relevant new approach methodologies (NAMs) that better recapitulate the peculiar nature of human physiology during pregnancy, especially the placenta and the maternal/fetal interface, which represent a key stage for human lifelong health. Fit-for-purpose NAMs for the placental–fetal interface are desirable to improve the biological knowledge of environmental exposure at the molecular level and to reduce the high cost, time and ethical impact of animal studies. This article reviews the state of the art on the available in vitro (placental, fetal and amniotic cell-based systems) and in silico NAMs of human relevance for developmental toxicity testing purposes; in addition, we considered available Adverse Outcome Pathways related to developmental toxicity. The OECD TG 414 for the identification and assessment of deleterious effects of prenatal exposure to chemicals on developing organisms will be discussed to delineate the regulatory context and to better debate what is missing and needed in the context of the Developmental Origins of Health and Disease hypothesis to significantly improve this sector. Starting from this analysis, the development of a novel human feto–placental organ-on-chip platform will be introduced as an innovative future alternative tool for developmental toxicity testing, considering possible implementation and validation strategies to overcome the limitation of the current animal studies and NAMs available in regulatory toxicology and in the biomedical field.

## 1. Introduction

Intrauterine life is the most sensitive life stage, and minimal perturbations may alter the natural programming of the developing fetus. An increasing concern has arisen in the last decades due to the evidence that exposure to chemical compounds during pregnancy may represent a main risk factor for the later onset of chronic adult diseases affecting millions of people worldwide, as described in the Developmental Origins of Health and Disease hypothesis [1]. Indeed, the use of developmental toxicants is strongly restricted by international laws, but a high number of compounds have to be yearly evaluated in Europe for their potential adverse effects on human health under the Regulation, Evaluation, Authorization and Restriction of Chemicals (REACH) regulation [2]. However, the actual risk-assessment process and the mode of action (MoA) evaluation are adding a delay in the decision making and may increase the disease burden due to continued exposure to chemicals.

Prenatal life is a very vulnerable state in which environmental exposures may trigger lifelong effects; the consequent disease burden may be very high and mostly unknown due to the multifactorial nature of the risk [3]. Taking endocrine disrupting chemicals (EDCs) as an example, the associated disease costs per year were estimated to be USD 340 billion in the USA, USD 217 billion in Europe and CAD 24.6 billion in Canada. These health economic costs represent, respectively, 2.33% of the US gross domestic product (GDP), 1.28% of the European GDP and 1.25% of the Canadian GDP [4,5]. However, these are greatly underestimated numbers for the burden of disease and considered by experts as a minimum, because calculations were performed “…only for chemicals and outcomes with a high probability of causation and for which exposure data were available” [6].

So far, toxicity testing of chemical compounds, including developmental toxicity, has been based on animal models, especially for regulatory purposes and risk assessment. However, interspecies differences and other limitations result in a high degree of uncertainty when transferring the results to human physiology, especially for developmental toxicity testing, due to placenta and maternal/fetal interface differences among species [7,8]. Further, this is no longer sustainable in the long run due to time-consuming procedures, high cost and the societal impact of animal studies [9]. 

Recently, the exponential growth of scientific and technological advancements prompted for a paradigm shift to integrate the use of human-relevant new approach methodologies (NAMs), including in vitro (e.g., human cell-based assays and systems), in chemico (cell free assays) and in silico (e.g., structure–activity relationship models) methods into risk assessment protocols. Indeed, several regulatory agencies already planned a progressive phasing out of animal studies and started to promote the use and integration of NAMs by scientists and risk assessors. Some international bodies have already defined and implemented a roadmap for NAMs inclusion in risk assessment, such as the US Strategic Roadmap [10] or the US Environmental Protection Agency’s (EPA) Strategic Plan [11]; in addition, active discussion on this issue is reported, for example, in a workshop organized by the European Chemicals Agency (ECHA) [12]. Very recently, the European Food Safety Authority (EFSA) launched a “roadmap for action” to integrate and implement the use of NAMs into risk assessment [13].

In order to set up a suitable and effective alternative method for developmental toxicity testing, aiming at replacing animal studies in the frame of the 3Rs’ principle, several issues have to be considered: (i) which are the endpoints of regulatory relevance requested by standardized test guidelines (TG) such as the Organization for Economic Co-operation and Development (OECD) TG 414 on Prenatal developmental toxicity study [14]; (ii) which are the available NAMs to assess developmental toxicity (i.e., embryonic stem cells (ESCs), fetal primary cells, induced pluripotent stem cells (iPSCs), etc.) if they are already fit-for-purpose with a biological relevance for humans in a mechanism-based context, or if they still need some tailor-made implementation; (iii) how the results could be translated for regulatory acceptance, e.g., by validation of the methods. 

Due to the complexity of the many possible adverse developmental effects and the current limitations of in vitro methods, it is unlikely that a single test could replace animal studies such as the OECD TG 414. Therefore, the development of an integrated system would represent a reliable solution, possibly in the framework of Integrated Approaches to Testing and Assessment (IATA) and Adverse Outcome Pathways (AOPs) approaches [15]. 

For a more realistic investigation of the possible alterations deranging physiological mechanisms related to developmental toxicity, cellular co-culture models should be employed, in particular those originating from different organs of the fetus which represent a relevant developmental stage. In addition, since a fetus grows surrounded by the protective and nurturing environment of placenta, amniotic fluid and membranes, in vitro models of these organs should also be integrated into a NAM aiming at identifying developmental toxicity effects of chemicals. Indeed, these organs represent the first barrier to external insults, themselves being subjected to exposure effects that may lead to disruption of proper placenta functionality, thus compromising proper fetus development. A step forward would be to integrate all these models using advanced approaches, such as ad hoc-implemented organ-on-chip (OoC) systems, strongly improving the predictive potential of developmental toxicity testing for humans. By such a platform, it would be possible to take into account a number of variables associated with human development, thus more robustly supporting the definition of adverse effects and MoA but also regulatory relevant data such as reference points for risk assessment.

With the ambition to fully replace animal studies in the future, the proposed model should also consider the implementation of in vitro cellular models representing both sexes at fetal level. Indeed, sex is a critical variable when studying adverse effects and MoA of chemicals, especially EDCs, and more importantly, during the development when the phenotypic sex is being defined. The same should be pursued also for AOPs, since currently most of the AOPs describing developmental adverse outcomes are not sex/gender-specific, while others are described only for one sex. 

In this scenario, the present article aims to analyze the state of the art on available in vitro (e.g., placental, fetal and amniotic) and in silico NAMs of human relevance for developmental toxicity testing purposes, as well as available AOPs related to endpoints relevant at regulatory level. Considerations on what is missing and needed will be discussed. Finally, a proposal of a novel human feto–placental OoC platform will be presented as an innovative alternative tool for developmental toxicity testing, discussing possible implementation, extended applications and validation strategies.

## 2. The Regulatory Context: The OECD TG 414

The gold standard for the identification and assessment of deleterious effects of prenatal exposure to chemicals on developing organisms is the OECD TG 414: Prenatal developmental toxicity study [14]. This TG has equivalents in protocol 870.3700 for USEPA and protocol B.31 for European regulations [16].

The assessment of developmental toxicity via OECD TG 414 is required for authorization of use of a lot of chemicals that human beings are going to be exposed to. Indeed, the REACH regulation demands that the OECD TG 414 test should be performed in at least one species for all substances with a tonnage band higher than 100 tons/year. Moreover, other substances outside REACH regulation, such as biocides or plant protection products, also require this developmental toxicity test. 

Although other OECD TGs include testing for developmental toxicity, such as the OECD TG 421: Reproduction/developmental toxicity screening test [17] and the OECD TG 422: Combined repeated dose toxicity study with reproduction/developmental toxicity screening test [16,18], they could only be relevant in terms of preliminary hazard identification and for setting priorities for further test requirements but not for detailed and reliable risk assessment due to their limited ability to unmask teratogenicity [19].

The main animal species used in the TG 414 is rat (Rattus norvegicus) in case of rodents and rabbit (Oryctolagus cuniculus) in case of non-rodents. Nevertheless, other animal species can be occasionally used when supported by robust scientific reasons, as is the case of mouse (Mus musculus) or Guinea pig (Cavia Porcellus).

The current version of TG 414 was launched in June 2018 when some endpoints for detecting EDCs were added, namely the anogenital distance (AGD) in fetuses and thyroid hormones in dams. Indeed, these are the only requested observations linked to a MoA, even if they are mandatory only for rodents but not for rabbits. All the other endpoints to be assessed in the TG 414 are related to general toxicity, implying gross pathological and histological observations as summarized in Table 1. Of note, no endpoint related to placenta is considered in the TG, except for uterine weight. 

The TG 414 requires that the test chemical is dosed on at least 20 pregnant females from implantation day one to as close as possible to the normal day of delivery. This period typically covers gestation days 5–15 in rats and gestation days 6–18 in rabbits. The chemical is orally administered, typically by gavage, although in some cases dermal and inhalation routes can be considered. At least three doses should be considered, plus the concurrent control, ideally spaced from two- to fourfold intervals of concentrations, with the top dose inducing some minor maternal toxicity (such as a slight decrease in body weight or mild clinical signs) and the lowest with no maternal toxicity. The study with three doses is not considered necessary whether a limit dose of at least 1000 mg/kg body weight/day produces no developmental toxicity.

Obtained data should be reported including individual animal data and a summary of the raw data in tabular form.

As for other animal studies, the economic cost of a TG 414 study is quite high. A survey carried out in 28 Good Laboratory Practice facilities based in the European Union and Switzerland to estimate the price of regulatory toxicity studies demanded by REACH [20] concluded that the average price of an OECD TG 414 test was EUR 63,100 in rats and EUR 92,500 in rabbits. If we multiply these costs to the number of compounds to be assessed for regulatory acceptance under REACH regulation, ranging from 68 to 101,000 chemicals [9], over EUR 4 billion would be necessary at the lowest for rats.

As a further matter of concern, although the number of pregnant females initially exposed is relatively low, the number of pups can be high. Indeed, the estimated number of animals required for a single test may vary between 150 [21] and 784 rats or 560 rabbits [22], thus reaching over 10 million animals at best. The use of animals for biomedical studies is cause of social concern due to ethical reasons. Indeed, in Western countries, statistics suggest that around 50% of the population is open to animal experimentation, whereas the remaining 50% strongly refuses the use of animals for biomedical applications and research [23] independently of the social welfare that may result. The idea developed in this paper moves forward a notable reduction of animal experimentation for developmental toxicity testing, thus contributing to improve the social consideration of biomedical research among the population.

## 3. Human-Relevant NAMs for Developmental Toxicity Testing

### 3.1. Human-Relevant NAMs for Placenta Toxicology

One aspect sometimes overlooked in developmental toxicity testing is that fetuses are exposed to chemicals through their mothers; thus, women’s exposure, their metabolism and placental health are critical issues to be considered. As described above, in TG 414, no endpoint regarding placenta is included while its full functionality is essential for the correct fetus development. 

The placenta is a transient organ with a relevant role as a barrier protecting the fetus and as an exchanging surface for nutrients, gases and metabolites [24]. However, many chemicals and/or their metabolites might pass this barrier, with potential health consequences for a safe gestation. Further, placenta is metabolically competent, thus playing an active role in metabolizing and delivering the chemicals from the mother to the exposed embryo/fetus. Importantly, placenta transfer may substantially differ among species, both for anatomical and metabolic reasons, making results obtained in animal models hardly transferrable to humans [8]. Indeed, compared to the hemodichorial and hemotrichorial placentas of rabbits and rodents, respectively, the human placenta is hemomonochorial and consists of the syncytiotrophoblast, which is in direct contact to the maternal blood, a reduced number of cytotrophoblast cells, stroma cells and the endothelium of the fetal capillaries [8,25]. Further, placenta has endocrine activity, thus representing itself a target organ for EDCs [24].

For these reasons, it is necessary to use and implement human placenta models that more closely recapitulate placenta physiology, therefore representing elective candidates for maternal and developmental toxicity testing.

#### 3.1.1. Ex Vivo Dual Side Human Placenta Perfusion

The most advanced available model is the ex vivo human placenta perfusion, a unique model that allows biological and toxicological testing in an intact and vital human organ outside the body. The major aim of this method is to test the transfer of substances through the placenta, mostly from the mother to the fetus, but also vice versa. The placenta can be obtained immediately after delivery and installed in the perfusion system within an ischemia time of less than 30 min. Routinely, it can be maintained completely functional for 6 h, which can be assessed by numerous parameters [26]. The method was invented and established in the late 1960s [27]. Since then, especially the equipment, the standardization and the medium composition have improved. 

Previous and recent original studies have focused on the transfer of drugs, but the transfer of a variety of environmental chemicals, including EDCs, has also been evaluated. A recent review summarized the studies performed so far grouping the chemicals on the basis of their feto–maternal ratio transfer [28]. Such a parameter is generally obtained comparing the transfer rate with an internal quality marker for valid perfusion experiments such as antipyrine or creatinine that are spiked into the maternal circuit and have a specific transfer kinetic to the fetal side. Compounds such as 2,3,7,8-tetrachlorodibenzo-p-dioxin (TCDD), glucoronated bisphenol A (BPA), perfluorinated compounds (PFC), glyphosate, medium-high molecular weight polybrominated diphenyl ethers (e.g., BDE-99 and -209) and some phthalate metabolites have been classified as having low transfer rate, whereas free BPA, BDE-47, genistein and, most of all, parabens, benzo[a]pyrene and acrylamide, have high transfer rates [28]. Being a metabolically competent organ, placenta may detoxify or even toxify compounds passing the barrier; thus, it is highly relevant to assess which metabolites and parental compounds have passed on the fetal side, for example, by metabolomics analysis.

Of course, a lower transfer rate does not mean the compound is less toxic; rather, it may affect placenta functionality. Noteworthy, the placenta perfusion model also allows assessing the influence of external factors on physiological functions of the placenta such as transport enzymes or hormone production [29,30]. 

The large volume of the perfusion circuits allows frequent sampling (e.g., every 15 min for up to 6 h) for analysis of pH, glucose consumption and lactate production as metabolic controls and β-hCG release, the main pregnancy hormone, as a measure for functionality of hormone production in the syncytiotrophoblast. The concentration of the test substance with the questionable placenta transfer, its metabolites or secondary mediators can be analyzed at the same or another frequency in the circuits. At the end of perfusion, the perfused tissue can be used for any kind of further biochemical or microscopical investigation assessing endpoints of placenta functionality.

The advantage of ex vivo human placenta perfusion is that it is very near to the in vivo functionality of the organ, which allows highly reliable results. Especially compared to animal experiments, the relevance for human medicine and toxicology is far higher, as placenta structure, size and functions vary immensely between species. However, it has to be taken into account that the model uses term placenta, which may differ from first or second trimester placenta as regards metabolism, endocrine activity and placental layers’ development. A disadvantage of the model is the enormous time consumption to complete valid sets of experiments and the need of a well-trained team and well-established cooperation with the delivery room [31]. Indeed, ex vivo human placenta perfusion is still a challenging method that is difficult to apply and with a low rate of successful experiments, as the placentas often have defects not visible macroscopically and the performing scientists and technicians need an intensive training period.

#### 3.1.2. Human Placenta Ex Vivo Explant Cultures

A level down, to using the whole organ and in between the common usage of cell lines or organoids, is the establishment of tissue cultures by explanting freshly harvested compartments of assembled cells still integrated in their in vivo microenvironment [32,33]. 

The main challenge in this regard is the long-time culturing in order to be accepted as an equivalent alternative to animal testing. As mentioned above, human placenta is the most readily available human organ and can be transferred to in vitro culture within minimal time after birth, to keep the in vivo situation stable without major changes in temperature, nutrition or microenvironment. In the past, the first placental explants to study transport mechanisms were prepared as 0.5 mm slices and cultured over two hours [34]. Experimental improvements helped to prolong the duration of viable cultivation and to maintain placental characteristics, such as hormone production and villous morphology, up to 48 or 72 h but rarely longer [35]. Automatized sampling and long-term culture of human ex vivo explant cultures in different toxicological applications can be afforded by using, for example, a microfluidic system and TissGrid^®^ 3D culture scaffolds [36]. 

Compared to cell or organoid cultures, placental explants have the advantages of being original primary tissue containing different cell types morphologically assembled as in the in vivo microenvironment and consisting of the typical placental barrier formed by syncytiotrophoblasts, cytotrophoblast cells and the endothelium of the fetal capillaries. Moreover, all surrounding cell types, such as stroma and immune cells, including Hofbauer cells, are preserved and can be investigated in culture over a long time (up to 28 days), although the different cell types have different behavior and viability over time [37]. After a few days of regeneration, the syncytiotrophoblast re-establishes [38] and produces hormones such as estradiol, progesterone or β-hCG. By adding different substances or drugs, toxic effects can be observed much longer than by the ex vivo placenta perfusion. Still, the explant culture system is comparably cost-effective and a suitable system for translational research. 

Further research is needed to evaluate the intra- and inter-individual variations of placentas and explants; however, these variations represent an advantage in toxicological studies since they reflect the variety among human individuals. Placenta explant studies offer the option of substituting animal experiments not only for testing acute toxic effects but also for mid- to long-term reactions. Indeed, several toxicology studies have been performed based on placenta explants as listed in Table 2; further, a previous review article has summarized placenta explant studies describing toxic effects of dichlorodiphenyltrichloroethane (DDT), organotin, PFC and PBDEs which affect, among others, hormone and prostaglandin secretion [39].

### 3.2. Human-Relevant NAMs of Amniotic Membranes

The developing embryo/fetus, other than being protected and fed by placenta, is also surrounded by amniotic fluid and membrane, whose inclusion and consideration in developmental toxicity testing is mostly neglected. The human amniotic membrane is of embryonic origin, and it is the innermost of the fetal membranes. It is in direct contact with the amniotic fluid via one of its two cell populations, a layer of human amniotic membrane epithelial cells (hAEC). The second cell population, the human amniotic membrane mesenchymal stromal cells (hAMSC), is interspersed within a rich extracellular matrix layer [45]. Due to the continuous cross-talk with both placenta and fetus, exposure to chemicals of amniotic membranes may cause inflammation of fetal membranes with drastic effects on pregnancy and fetus development and with possible consequences later in life. Indeed, an aberrant release of cytokines can be toxic to organs such as brain, leading to preterm brain injury. Effects on isolated human and murine amnion epithelial cells have been described following exposure to: (1) tributyltin, which impairs the amniotic membrane through the disruption of tight junctions [46]; (2) polychlorinated biphenyl (PCB) 50 congener, which may lead to inflammation in fetal membranes [47]; (3) BDE-47 and -99, which may lead to induction of cellular senescence in vitro, possibly through oxidative stress [48]. 

It is relevant to set up reliable in vitro models of human amniotic membranes to understand the pathophysiology of many pregnancy and fetal complications induced by exposure to chemicals. However, the cell source to use as a model is still a matter of discussion. In fact, it was recently discovered that the transformed epithelial FL cell line was misidentified and indeed is a HeLa derivative (https://www.atcc.org/products/ccl-62, accessed on 18 August 2022); notwithstanding, they were often used and reported in the scientific literature, also for toxicology studies [49,50]. While considering human stem cells, the pluripotency potential of human cells derived from fetal annexes and human iPSCs has been compared, and differences at epigenetic level were highlighted [51]. Further, several attempts have been made to develop in vitro models represented by amniotic cells, explant cultures (floating or in transwell) [52], and strategies towards an amniotic membrane on a chip [53]. 

A substantial improvement has been obtained by separating the two relevant sub-regions of the human amniotic membrane, the placental and the reflected amnion, demonstrating that these areas respond to distinct sub-regional differences including metabolic and secretory parameters [54,55,56], thus possibly representing different targets displaying different outcomes following exposure to chemicals. Moreover, to overcome the loss of viability and functionality during cultivation of floating membrane biopsies, the system has been distended, obtaining an extension of the life span and the physiological function in vitro for up to 21 days [57]. This may represent the most promising amniotic membrane model whose implementation and adoption should be recommended for a useful in vitro tool to test the impact of environmental pollutants, such as EDCs or nanoparticles, on mitochondrial status, inflammatory status, senescence and apoptosis and parameters of membrane rupture, still poorly investigated.

### 3.3. Human-Relevant NAMs for Embryonic/Fetal Toxicology

The main goal for developmental toxicity testing is represented by the direct assessment of exposure effects of chemicals on the fetal/embryo development of organs in a precise window of gestation before birth. The TG 414 described above only considers the evaluation of macroscopic endpoints, but the application of alternative NAMs varying from morphological, to proliferation, metabolomics, proteomics, epigenomics and transcriptomics, till to functional endpoints related to the cell type and assay used will expand the range of endpoints, still complying with the relevant adverse outcomes of TG 414. 

Different types of NAMs involving chemical effect assessment on fetal/embryo systems have so far been explored as developmental toxicity testing. They mainly belong to three approaches, whose pros and cons are summarized in Table 3: (1a) in vitro cultures of differentiated ESC lines or (1b) iPSCs from humans [58,59,60,61] organized in 2D or 3D structures, spheroids and organoids; (2) in vitro systems using fetal/embryo primary cell/organ cultures [62].

Conceptually, both models allow dissecting the direct effects of toxicants on the isolated developing organs avoiding any systemic effect and interaction with other organs during embryogenesis. Only an ex vivo whole embryo culture would retain the systemic interactions between organs as well as better resemble the way of toxicant exposure. Since an ex vivo human embryo system cannot be set up for evident ethical reasons, a zebrafish embryo teratogenicity assay has recently been established as an alternative method to predict human developmental toxicity as a whole organism [63].

#### 3.3.1. In Vitro Cultures of ESCs and iPSCs and Derived 3D Organotypic Models

Human embryonic development covers the first 8 weeks of gestation. hESCs are pluripotent cells that can be isolated from the inner cell mass of the blastocyst before implantation, retaining unlimited ability of self-renewal and of differentiating into all embryonic tissues and extra-embryonic annexes. Since their isolation causes the destruction of the blastocyst, their use is strictly regulated and impaired by ethical and clinical concerns; moreover, in vitro cultures may result in teratomas. To overcome these restraints, iPSCs have been developed by reprogramming adult somatic cells to a pluripotent stage through retroviral transduction of specific genes according to the Yamanaka’s protocols [64]. 

The majority of the studies adopting in vitro cell systems based on hESCs and hiPSCs have been focused on neuro- and cardio-toxicology [65,66,67]. For the developing brain and heart, where the complexity of the cellular composition and of the spatial organization of the organ can affect the functionality as well as the sensitivity to toxicant insults, developmental toxicology research has incredibly benefitted from the establishment of 3D organized structures called spheroids or organoids [66,67]. Three-dimensional gastruloids, aggregates of embryonic stem cells that recapitulate key aspects of gastrula-stage embryos, have emerged as a powerful tool to study early stages post-implantation development in vitro [68,69]. As gastrulation is the checkpoint for the body formation plan during embryogenesis [70], this new 3D model is rapidly affirming as a validated model to test developmental toxicity of chemical exposure [61].

Although promising, these iPSC-based assays have not yet become the gold standard for developmental toxicity testing. Indeed, assessment of the adverse impact of new chemicals on embryogenesis by these approaches is not yet recommended by regulatory agencies. Reports from the International Council for Harmonization’s guideline on Detection of Toxicity to Reproduction for Human Pharmaceuticals generally recommend the use of in vitro assays for developmental toxicity assessment but with no endorsement of any specific stem cell assay [71]. Thus, validation of the application of these models for developmental toxicity testing is still far from being achieved, a condition that makes stakeholders, including regulatory agencies and chemical companies, not yet able to adopt any specific stem cell-based assays as part of the decision-making process in developmental toxicity assessment.

The best in vitro assays using iPSCs should define the inhibitory impact of the chemical exposure on viability, differentiation or cell functions as a sign of developmental toxicity. Many types of developmental toxicity assays have been reported until now for ESC and iPSCs, based on the ability to differentiate towards a variety of cell types, including neurons and osteoblasts [72]. Morphological and molecular techniques to measure the adverse impact on cell differentiation are used to constantly improve the sensitivity and specificity of the test.

One of the typical morphological endpoints for developmental neurotoxicity (DNT) is the formation of rosette structures that are an in vitro indicator of the successful closure of the neural tube during development [73]. Moreover, demyelination represents a relevant functional DNT endpoint, as axon myelination in the central nervous system starts early during the first months of embryogenesis and is one of the key events of brain development [74]. Some in vitro human neural models have been developed for DNT studies, as comprehensively reviewed elsewhere [75,76]. The most challenging aspect for these models is to reproduce in vitro cell population heterogeneity (neurons, astrocytes, oligodendrocytes and microglia) in a 3D cytoarchitecture that can mimic the brain’s physiological activity and in the case of DNT studies, the specific developmental stage of exposure. hiPSCs can be functionally differentiated into cell lineages organized in organoids comparable to brain development in utero [77]. Oligodendrocyte spheroids and oligodendrocyte precursors have been obtained from iPSCs for specifically studying the myelination process [77,78,79,80], while 3D models have been generated from hESCs reproducing in vitro brain structures consisting of mixed populations of neurons, astrocytes and oligodendrocytes with immature myelin sheaths [81]. 

An innovative 3D model of human brain (mini brains) has recently been developed from human iPSCs [82]. These organoids, named BrainSpheres, can recapitulate several morphologic, architectural and functional aspects of the human brain as they are formed by a mixed population of neurons, astrocytes and oligodendrocytes with high level of myelinization and spontaneous electrical activity; for these characteristics, they are extensively used for DNT studies and neurodegenerative diseases [83,84,85]. This model is particularly interesting as it can also be co-cultured in a transwell system with the endothelial cell barrier to test the differential penetration ability of the toxicants [86]. Moreover, it has recently been used to perform the first developmental toxicology screening for reference compounds on the myelination process of oligodendrocyte differentiation, based on an extensive search of the neurodevelopmental toxicity literature [85]. 

The PluriBeat assay has been set up to test cardiotoxicity by developing 3D cultures of human iPSCs (hiPSCs), termed embryoid bodies, which model the human blastocyst and can be differentiated into beating cardiomyocytes undergoing molecular events similar to those of the developing embryo [87]. More recently, in vitro mixed cell systems for assessing chemical toxicity on mesoderm and endoderm-derived organs during embryo development have been established [72] as highlighted in the last update of the DevTox database [88], extending the horizon of in vitro developmental toxicity testing. For an updated and exhaustive analysis of the methods and type of tests adopted, see Mennen et al., 2022 [72].

A pivotal aspect that should be considered when working with differentiated cells from iPSCs or ESCs is that there could be a temporal and developmental gap between the differentiated cell system and the in vivo developmental window sensitive to adverse effects exerted by the chemicals to be tested. Indeed, the protocols of reprogramming and differentiation are currently not sophisticated enough to allow differentiation to a temporal extent of embryogenesis [64,89,90]. In addition, it is current opinion that each organ and axis, in particular the endocrine ones, have a specific temporal window of sensitivity to chemicals that accounts for any difference in the toxic effects measured. These sensitive windows also differ between humans and other animal models [91] and may also be affected by genetic sex as influenced by sex hormone production. Thus, in vitro differentiated ESCs or iPSCs may not reflect the sensitivity stage of development, thus potentially affecting the validity of the obtained toxicity results. Moreover, when lineage differentiation is induced in vitro, the differentiation potential may be lost or altered, resulting in limited effectiveness of the readouts used to assess toxicant effects on the differentiation potential of the cells. Some examples of the use of hESCs for evaluating toxicant effects of different chemicals on specific induced lineages are summarized in Table 4.

#### 3.3.2. Primary Fetal Cell Culture Models

Fetal primary cell cultures from specific tissues or organs have been obtained from abortive material (weeks 9–12 of gestation) and can be cultured in vitro in monolayer for a longer period compared to 3D organoids without losing their properties (functional activities, differentiation potential, stem and proliferative properties), their phenotypic, as well as their genetic profile. Since the cells are isolated from waste abortive material, their use raises no major ethical concerns; thus, they could represent promising and valuable in vitro systems for developmental toxicity testing. However, until now, they have mainly been used in pharmacological studies and not yet validated for toxicity screening at a regulatory level. 

Several different fetal cell populations derived from the developing central nervous system and neural crest have been characterized so far. Neuroblast cell populations (FNC-B4) have been derived from a fetal olfactory epithelium and extensively characterized in in vitro cultures to respond to sex steroids, kisspeptin and odorants by secreting gonadotropin-releasing hormone in both physiologic and pathophysiologic conditions [102,103,104,105,106]. They markedly respond to estrogens, androgens and thyroid hormones [107,108,109], thus making them a potentially relevant endocrine model to assess developmental effects of EDCs, as recently shown for the widespread persistent organic pollutant Benzo[a]pyrene [99]. To study cholinergic response in human developing central nervous system, a population of immature cholinergic neurons has been characterized from the Nucleus Basalis of Meynert in the basal forebrain [110]. 

A neuroendocrine population has been isolated from the trunk region of the neural crest of human fetuses at the edge of the forming neural tube. Their secretory and expression profile confirmed they retained in vitro the ability to differentiate toward the sympatho-adrenal phenotype, reflecting their in vivo destiny to migrate and populate the medullary catecholamine-secreting core of the adrenal [111]. 

Inducible inflammatory response and active cytokine secretory activity have been characterized in fetal cardiomyocytes [112] and skeletal muscle cells [113].

Finally, the establishment of human smooth muscle cell cultures from fetal male external genitalia enabled us to study the androgen and estrogen effects on cell proliferation and contractility [114], being also implied to assess adverse effects of PCBs as risk factors of developmental external genitalia diseases such as hypo- and epispadias [100]. 

To complete the scenario of the current available in vitro primary human fetal cell models that may be applied to developmental toxicological studies, a cell model of adipose fetal precursor retaining in vitro the adipogenic potential versus brown adipocytes was established [115]. This model could represent a unique system to investigate the developmental basis of metabolic pathologies such as obesity, lipodystrophy and diabetes. 

These cell models, together with the one obtained from fetal adrenals [116], enrich the fetal adrenal cell systems to be used for investigating in vitro the possible prenatal alterations that may result in the development of chronic pathologies later in adult life [1].

#### 3.3.3. Organotypic Fetal Models

Organotypic culture models can also be obtained by ex vivo maintenance of tissue or organ explants obtained during embryo development or can consist of 3D spheroids/organoids cultured in vitro and derived from spatial aggregation of cell populations derived from human fetal material. In both cases, organs or cells can be isolated from abortive material discarded in elective legal termination of pregnancy during the first trimester of pregnancy. 

In the 3D organization, cell–cell spatial interactions are maintained, thus representing more realistically the physiological conditions compared to conventional cell cultures and being more suitable to test the effects of toxicants on embryo/fetus development. Valuable examples of these models have recently been obtained for developing human adrenals starting from fetal explants [117], or as in vitro-induced organoids derived from primary cells isolated from fetal adrenals [116]. Both 3D models can be cultured in vitro in static conditions (no microfluidic system) up to 10–15 days and maintain functional endocrine activity without gross degeneration, before necrotic processes start. Of note, the organoid (called adrenoid) spontaneously formed from the mixed cell population obtained from dissociation of the fetal adrenal display a spatial organization and a functional activity (steroidogenic and catecholamine production) resembling the adrenal organ of origin [116]. 

The effect of steroidogenic inhibition on steroid hormone production in basal conditions and under Adreno Corticotropic Hormone stimulation has been studied on fetal adrenal organotypic culture [118]. 

A 3D model derived from commercially available fetal human neural progenitor cells (hNPCs) induced in vitro to microsphere has been successfully used to assess the endocrine disrupting effects of PBDEs on oligodendrocyte differentiation [101].

Testis and ovary organotypic cultures have also been established to assess the effect of manipulation of fibroblast growth factor 9 signaling on sex-specific gonadal differentiation, demonstrating that this grow factor is fundamental for the correct developing program for both the germ cell and the steroidogenic compartment [119].

#### 3.3.4. Remarks

These in vitro experimental approaches, which reduce the complex biological system of the embryo/fetus development to simplified cell or tissue systems, display the advantage of facilitating quantitative measurement of the specific responses to chemical perturbations [120]. However, they have the disadvantage of lacking the cellular/organ interaction and spatial dynamics playing a relevant role in the potential effects of toxicant exposure. Thus, integration of these types of data and their elaboration by computer modeling is mandatory in order to reconstruct such a complexity (as described further on). Further, a single type of assay and readout in vitro, based on a limited representation of embryogenesis and fetus development, would not be able to detect a whole range of chemicals and their mixtures that probably target different embryological steps with different mechanisms. Therefore, the standardization and validation of readouts and assays is pivotal, as well as the careful selection of multiple types of assays to be combined, in particular if developed considering available AOPs in an IATA framework. The interpretation of the results obtained by a testing system that is independent from the whole-embryo model and relies on focusing on specific parts should be carefully evaluated to be extrapolated to the in vivo situation.

### 3.4. Immortalization of Cellular Models in Developmental Toxicology Testing

Among the characteristics identifying the different cellular models described thus far, there are cell growth and cultivation lifetime, which are critical issues when several developmental toxicity testing are carried out. 

As described above, primary cells can be isolated from various tissues of the organisms, and protocols have been established to cultivate many different cell types in vitro. These cells closely reflect the properties of cells in vivo. However, in primary cells, cell growth is tightly controlled by the tumor suppressor genes p53 and the retinoblastoma protein (pRB) [121] and further influenced by the progressive telomere shortening occurring during every replication cycle [122]. Therefore, cell numbers required for industrial or high throughput applications are difficult to obtain using cellular assays based on primary cells. Although primary fetal cells tend to display a higher proliferation capacity compared to their adult counterparts [123], their use in industrial settings may raise ethical concerns, therefore limiting a widespread use.

Standard cell lines do provide the ease of use needed for screening tens of thousands of compounds in a high-throughput manner, due to their unlimited cell proliferation which is achieved by inactivating the above-mentioned safeguard mechanisms. Immortalization can be achieved spontaneously also in primary cells by specific cultivation regimen [124]; however, this method is cumbersome and is only applicable to certain cell types and very inefficiently. The introduction of viral oncogenes such as, e.g., SV40 large T antigen [125], human papilloma virus E6 and E7 genes [126] or adenoviral E1A and E1B genes [127], increased the immortalization efficiency dramatically; however, the drawback is that these genes very often induce genomic instability and thus lead to an altered karyotype in the resulting cell lines. A milder immortalization regimen was achieved when the catalytic subunit of the human telomerase was introduced into human primary cells [128]. In this case, the recombinant hTert maintains the ends of the telomeres, which would otherwise shorten with every DNA replication/cell doubling. As this progressive telomere shortening is circumvented by the ectopic expression of hTert, the primary cells become immortal and, most importantly, maintain functionality [129,130] and often also chromosomal integrity. Although very promising, especially for the expansion of fetal cells [131], the immortalization of hTert regimen is not universally applicable. It rather seems that, especially for adult or for fully differentiated cell types, the activities of additional oncogenes are required [132,133]. 

In addition to these universally acting immortalization or expansion genes, reports have also shown that specific genes preferably expand or immortalize only restricted cell types such as, e.g., v-myc for macrophages [134], Id1 for keratinocytes [135], Bcl-xl and Bcl6 for B-cells [136], to name just a few. 

Based on these observations, a completely new approach was established which was inspired by that adopted by Yamanaka to identify the iPSC-generating genes [64]. In this setting, a small library of genes was used to identify the most promising gene combination for the immortalization of different cell types. Thereby, not only gene combinations could be identified that lead to the immortalization of the respective cell types, but also these novel established cell lines maintain the cell physiology of the primary cells they were derived from [137]. This approach was used to generate novel physiologically relevant cell systems from human and animal species such as endothelial cells [138], alveolar and airway epithelial cells [139,140,141], osteoblasts [142], mesenchymal stem cells [143], placental cells [144] and thyrocytes [145].

As mentioned above, other approaches such as the in vitro differentiation of iPSCs/ESCs into the desired cell type, or the most recent advancement of organoids, allow the generation of physiologically relevant cells in sufficient quantities. The technologies, advances and potentials have been nicely and comprehensively reviewed elsewhere [146,147]. The major disadvantage of these approaches is that the generation of large numbers of differentiated iPSCs and organoids is extremely expensive, time-consuming, requires sophisticated protocols and thus represents a significant hurdle for the use of such cell systems in routine applications such as developmental toxicity screenings.

### 3.5. In Silico Models for Developmental Toxicity

In silico modeling is a board term that comprises several methods relying on computer simulations to estimate or predict different kind of properties. Among them, quantitative structure–activity relationships (QSAR) and read-across are often used to estimate toxicological properties of chemicals from their molecular structure or similar analogues, respectively [148]. Some attempts have been made to use these techniques as a kind of full replacement of the test itself, as it was performed for other quite complex endpoints such as carcinogenicity, so the apical effect is often modeled in such a way that they can be considered non-testing methods [149]. 

There are currently some QSAR models addressing developmental and reproductive toxicity that have been summarized in a recent review [150]. In general, these models depend on the availability of collections of data for a specific endpoint. In the case of the experimental data, according to the OECD TG 414, the number of substances evaluated is quite limited, thus reducing the effectiveness and capability to develop robust and reliable models. The most common strategy adopted to overcome the data limitation is to develop classification models, meaning that original data are often labelled as toxicant or non-toxicant. This is because it is easier to obtain classification rather than quantitative-predicting models and allows pulling together different data sources. At the same time, the utility of models predicting a toxic or non-toxic label is more limited compared to the original ambition of replacing an experimental test since this kind of information can be useful for screening purposes but not enough to be used for regulatory purposes (such as for the general adaptations allowed in REACH legislation for dossier preparation). 

One of the available classifiers for developmental toxicity is the CAESAR one [151], based on a collection of data for about 200 substances that have been evaluated by experts and labelled as positive or negative according to a risk–benefit analysis on their use during pregnancy. The model is based on a set of molecular descriptors and a random forest as algorithm, and it is publicly available within the VEGAHUB platform (www.vegahub.eu, accessed on 18 August 2022). A quite similar model is also available within the USEPA TEST platform (https://www.epa.gov/chemical-research/toxicity-estimation-software-tool-test, accessed on 18 August 2022).

An analogous study was performed by Wu et al. [152] at Procter & Gamble, and the developed model was made available through VEGA and the OECD QSAR Toolbox (https://www.oecd.org/chemicalsafety/risk-assessment/oecd-qsar-toolbox.htm, accessed on 18 August 2022) as a profiler. It covers not only developmental but also reproductive toxicity (DART). The authors collected and critically assessed evidence from the literature about DART effects of chemicals. Compared to CAESAR, which is a machine-learning model, this model is an expert system, since it implements a large collection of rules that have been developed by experts by analyzing the common moieties of toxic compounds and associating them to known mechanisms of developmental and reproductive toxicity such as binders of the retinoic acid receptor. The model considers some tens of skeletons, each of them representing a family of toxicants and, for each skeleton, a number of identified substituents counting more than 180,000 possible substances generated by the different alternative substituents. In the VEGA implementation, a substance is assigned as positive for developmental and/or reproductive toxicity, while the Toolbox profiler flags the compound as associated or not to previously known toxicants. 

In our experience, being the dataset at the basis of this model skewed for a large presence of toxic compounds, the model is often over-conservative, and the reliability associated to non-toxic predictions (as indicated by the applicability domain) is often low due to the poor representativeness of non-toxic compounds. Therefore, the user should consider this model as a profiler such as in the Toolbox, indicating a possible activity or common clustering features and not for a precise prediction. 

Other authors developed their models more linked to animal studies, not necessarily related to OECD TG 414 but also considering higher-tier tests. Classification models have been developed according to the following types of effects: male reproductive toxicity, female reproductive toxicity, fetal dysmorphogenesis, functional toxicity, mortality, growth and newborn behavioral toxicity [153,154]. Authors used the MC4PC algorithm, also called MultiCASE, depending on the versions, identifying fragments (also called toxicophores) associated to the toxic effect. These MC4PC models have lower sensitivity and higher specificity; thus, they behave opposite compared to the Procter & Gamble model. These kinds of models can provide information closer to the animal experiment but still missing dose response information useful for risk assessment.

More recently, interesting modeling efforts explored the combination of chemical information (hundreds of chemical fragments) with biological activity extracted from nearly 2000 toxicological high-throughput screening assays extracted from PubChem and ToxCast [155]. The inputs were used to group assays based on their chemical–mechanistic relationships and identified two clusters where the in vitro assays were enriched with developmental toxic compounds.

Some studies compared the results from different models and discussed in detail the use and interpretation of the results [156,157,158] sometimes applied to specific chemical classes. Overall, as a rule of thumb, the integration of different in silico models may improve the results [159,160,161].

Some available interesting models link developmental toxicity and endocrine disruption [162,163,164], whereas others are specific for endocrine disruption, but we will not address them here. Read-across approaches are also used to assess the potential developmental toxicity of chemicals by comparing structure and toxicological features of one or few substances being quite similar to the target compound, thus obtaining a final assessment on the basis of the weight-of-evidence [165]. This latter method was more frequently employed to fulfill standard requirements for developmental toxicity in the dossiers submitted until 2019 for REACH, being chosen for 30% of the studies (while less than 1% used QSAR information) according to the ECHA report for the use of alternatives to animal testing [166]. A limitation of read-across is that it relies on already existing data, somehow limiting its applicability.

A one-to-one replacement of an animal testing with a NAM is often not feasible for very complex endpoints. Therefore, an interesting contribution of in silico methods is in their support to experimental in vitro NAMs to help in covering the gaps compared to in vivo experiments. Indeed, QSAR models and other computational models such as PBK (Pharmacologically Based Kinetic) models can be used to better inform on the distribution process in a living organism and to cover those processes not fully represented in the in vitro system. For instance, QSAR models investigated the placental permeability [167,168] or the transfer of chemicals from the mother to the fetus [169,170,171]. Several studies focused on the computational assessment of metabolism, or other ADME properties were recently reviewed in [172,173].

As regards computational modeling applied to cellular models, an interesting open-source platform of cell–cell interaction modeling has recently been created by NIH and EPA (http://www.compucell3d.org, accessed on 18 August 2022). Different computer dynamic simulations of processes occurring during different phases of embryo development have been obtained using this platform, including urethral fusion during sexual diversification of the genital tubercle [174] and fusion of the secondary palatal processes [175]. Applying information derived from toxicant exposure to these models would be precious for a further implementation of in silico modeling and predictions of developmental toxicology.

Recently, a predictive virtual embryo represents the ultimate effort of computational modeling to represent human embryogenesis and assess the effects of perturbations resulting from toxicant exposures on human development (https://www.epa.gov/chemical-research/virtual-tissue-models-predicting-how-chemicals-impact-development, accessed on 18 August 2022) [175].

Overall, in silico methods can assist different steps of developmental toxicity testing, from the classification and identification of stressors to the simulation of in vitro or in vivo experiments, thus representing highly valuable tools to be integrated in developmental toxicity NAMs and IATA approaches.

## 4. Available AOPs Related to TG 414

In 2012, OECD introduced a new concept framework to gather and organize available evidence in a modular format [176]. Through this approach, it is possible to link perturbations determined by any exogenous stimulus (chemical, physical or microbiological) at molecular level, named the Molecular Initiating Event (MIE), to an adverse outcome (AO) in an organ, organism or the whole species, through a series of related and sequentially dependent key events (KE) (Figure 1).

The AOs should be of regulatory relevance; thus, their adoption by risk assessors is being increasingly considered. Once defined, the AOP prompts the development of a battery of assays/approaches linked to each KE, quantitatively providing the weight of evidence for each key event relationship (KER), thus falling into the IATA framework [15].

In a developmental toxicology context, the availability of AOPs describing pathways connected to apical endpoints, possibly related to the OECD TG 414, is of great relevance. With this aim, we searched among the 379 AOPs proposed so far, completed or under development and freely accessible through the AOP Wiki Portal (https://aopwiki.org/, accessed on 5 May 2022), those in which the «Life Stage applicability» domain corresponded to “Conception to fetal”, “Fetal to parturition”, “Development”, “Embryo”, “Fetal”, “Perinatal”, “During brain development” and “Pregnancy”, also including “All life stages” to check if the developmental stage was considered in the description. For the purpose of this review, only AOPs with a «Taxonomy applicability» domain listing mammalian species (i.e., human, monkey, rat, mouse) were included, thus discarding amphibian and aquatic species. The AOPs not reporting these two applicability domains were excluded, although some of them could have been relevant. Sixty-seven AOPs met the criteria, but thirteen were further excluded for being related to SARS-CoV-2 infection, external stimuli as light and ionizing radiations, inhalation or dermal contact as route of exposure, thus out of the scope of the present paper. Moreover, other seven AOPs were removed as being “Not under active development” or “Open for adoption”; thus, it is not guaranteed they will be completed. As a result, 47 out of 379 available AOPs were considered and grouped on the basis of the organ system affected in the AO (Appendix A).

### 4.1. Neurodevelopmental Toxicity

Among the nine AOPs related to DNT (Appendix A), four share “impairment of learning and memory” as AO. The corresponding MIEs include binding of antagonist compounds to N-methyl-D-aspartate receptors (NMDARs) (AOPs 12 and 13), binding of electrophilic chemicals to SH-/selenoproteins (AOP 17) and inhibition of Na+/I− symporter (NIS) (AOP 54). All these AOPs, except AOP 17, are completed and endorsed by OECD, being also inter-related in an AOP network on neurotoxicity [177,178,179,180]. Indeed, both NMDAR and NIS inhibition lead to the reduction of brain-derived neurotrophic factor (BDNF) release through, respectively, the decrease of intracellular calcium levels or of thyroid hormone (TH) synthesis and, consequently, T4 in serum. Loss of BDNF causes a drop of synaptogenesis with repercussions on neuronal network functionality and, ultimately, on learning and memory.

Disruption of TH homeostasis is also crucially involved in AOPs having “Decreased cognitive function” as AO (AOPs 42, 134, 152 and 300), with different MIEs but all inducing a decrease in T4 serum levels, thus affecting gene expression and physiology of hippocampus [181]. In addition, with the same AO but through a different mechanism, AOP 405 starts with acetylcholinesterase (AChE) inhibition as MIE.

Overall, AOPs describing DNT are limited to one brain area only, not taking into account possible sex-specific effects. Thus, the development of new AOPs related to AO occurring in other brain areas, especially those being sexually dimorphic, is needed.

### 4.2. Reproductive Toxicity

Regulatory relevant AOs of the male reproductive tract are described by ten AOPs (Appendix A) all involving, as KEs, a decrease of testosterone levels (mainly in Leydig cells) or of androgen receptor (AR) activation. Both KEs determine an altered (feminized) or incomplete development of male organs (AOPs 18 and 124), cryptorchidism (AOP 288) or short AGD (AOPs 305, 306, 307), as well as decreased sperm quality in adulthood (AOPs 66, 67, 68, 74), all conditions severely affecting fertility. Different MIEs may trigger these outcomes, including activation of some nuclear receptors unbalancing testosterone synthesis (AOPs 18, 66 and 67), binding of antagonist compounds to AR (AOP 306), inhibition of enzymes involved in the steroidogenesis (AOPs 124, 288, 305 and 307), as well as alteration of proteome or hypermethylation in fetal testis (AOPs 68 and 74).

Only two AOPs are related to the female reproductive tract. However, in AOP 167, the MIE is the increase of ER activity in the pre-pubertal phase, even if the “Life stage applicability” domain comprises the “Fetal to Parturition” in addition to the “Juvenile” term; thus, it appears that this AOP could not be strictly considered relevant for developmental toxicology. On the contrary, AOP 398 links the inhibition of the Aldehyde dehydrogenase 1A enzyme during the fetal life (MIE) to a reduction in follicle pool in ovaries, leading to decreased female fertility in adulthood.

While AOs related to male reproductive toxicity are substantially covered by the current AOPs, including AGD shortage considered in the OECD TG 414, very little information is available for females. Indeed, in the frame of the TG 414, an AOP on female alterations of AGD would complete the information on this endpoint. In addition, AOPs related to uterus are missing. Thus, the implementation of AOPs on developmental reproductive adverse effects occurring in sexual organs, especially in females, should be encouraged.

### 4.3. Cardiovascular Toxicity

Among the nine AOPs considering cardiovascular toxicity endpoints, only three are strictly related to development, while the others may occur in all life stages, with limited relevance to fetal life (Appendix A).

Cardiovascular developmental toxicity is mainly represented by the impairment of the vascular endothelial growth factor (VEGF) pathway, considered in two AOPs having as MIE the disruption of VEGF receptor (VEGFR) signaling (AOP 43) [182] or the activation of AhR (AOP 150, completed and endorsed) [183]. Both MIEs lead to a reduction in angiogenesis and impairment of the endothelial network and culminate in developmental defects and early life mortality, respectively. In addition, AOP 436 describes the inhibition of retinaldehyde dehydrogenase leading to altered positioning of the great arteries in heart.

Among the AOPs having “All life stages” as Life stage domain, AOP 16 describes the inhibition of AChE, which determines both cardiovascular and respiratory impairment with consequent mortality. Other AOPs imply the impairment of ion channels leading to increased cardiac arrhythmia and mortality (AOP 104) or calcium channel blockade till heart failure (AOP 261) or inhibition of hERG potassium channels, which determines cardiac arrhythmia (torsade de pointes) and sudden death (AOP 433).

Since alterations in the cardiovascular system are considered only as gross malformations in the OECD TG 414, these AOPs, especially AOPs 43 and 150, may be of support in developmental studies using alternative testing methods [184].

### 4.4. Liver Toxicity

Among the four selected AOPs describing liver toxicity (Appendix A), only AOP 46 reports “During development” in the Life stage domain and describes the mutagenic action of Aflatoxin B1 causing hepatocellular carcinoma. AOP 107 has the same AO, but it is triggered by the activation of the constitutive androstane receptor. Moreover, in AOP 220, completed and endorsed, the AO is liver cancer, triggered by the prolonged activation of the CYP2E1 enzyme [185]. 

AOP 209 has no classified MIE, starting with two KEs, namely the activation of sterol regulatory element binding transcription factor 2 and inhibition of glutathione synthetase and glutathione S-transferases as KEs, determining perturbation of cholesterol and glutathione homeostasis, respectively, and contributing to hepatotoxicity. 

Since only one AOP on liver toxicity is specifically linked to the developmental phase, it would be advisable to develop more AOPs focused on this window of exposure, also considering possible alternative MIEs and AOs, such as, for example, alteration of alfa fetoprotein or other CYP levels, also taking into account the sexually dimorphic function of the liver.

### 4.5. Other Soft Organs’ Toxicity

AOPs describing AOs in other soft organs have not been specifically designed for the developmental stage. In particular, there are two AOPs related to lungs, and one for kidneys, stomach and pancreas (Appendix A). The ones on lung toxicity (IDs 39 and 206) imply an increase of inflammation processes leading to allergic respiratory hypersensitivity and to lung fibrosis, respectively. Compared to other AOPs in lungs excluded for being related to the inhalation route, these were kept as potentially representing critical pathways also for developmental lung toxicity.

AOP 257 describes kidney toxicity induced by endocytosis and lysosomal overload. Although drugs are mostly included as stressors, we kept this AOP since also cadmium, an EDC, is present.

Both AOPs in pancreas and stomach are related to cancer induction. While pancreatic acinar cell tumor may occur in fetuses (AOP 316), the treatment-resistant gastric cancer (AOP 298) appears more related to adult life; thus, it should be considered with caution in the context of developmental toxicity. 

Although considering alterations of relevant signaling pathways, the applicability of these AOPs to developmental toxicity should be carefully evaluated.

### 4.6. Skeletal Malformations

The only AOP in this domain that fulfils the inclusion criteria describes an AO in the skeletal system, which is also considered in the OECD TG 414 (Appendix A). Although authors considered anti-epileptic and anti-arrhythmic drugs as stressors triggering the inhibition of cardiac voltage-gated sodium channels, other chemical compounds may affect this pathway, determining skeletal malformations, including amputations.

### 4.7. Immunotoxicity

Three AOPs describe immunotoxicity applicable to “All life stages” (Appendix A). In two of them (AOPs 154 and 315), the impaired T-cell dependent antibody response AO is, respectively, triggered by inhibition of calcineurin activity or of Jak/Stat signaling. The third AOP links estrogen receptor α binding in immune cells to the exacerbation of systemic lupus erythematosus. This AO seems more relevant for female health, especially during pregnancy, and it may represent a risk factor for fetuses. Thus, it would appear appropriate to group it with AOPs in pregnancy, described below.

### 4.8. Other Developmental Outcomes

AOP 202 describes the occurrence of infant leukemia following binding to topoisomerase II, which occurs early during development (Appendix A). Two other AOPs (IDs 263 and 265) share the same AO, growth inhibition, as well as the same MIE, i.e., uncoupling of oxidative phosphorylation, but implying a decrease in cell proliferation and in lipid storage as KEs, respectively. Increased oxidative DNA damage is instead the MIE of AOP 296, leading to increased mutations and chromosomal aberrations.

### 4.9. Placental Toxicity

Only two AOPs are available in the AOP-Wiki portal pertaining to maternal toxicity during pregnancy (Appendix A). Of them, only AOP 151 is directly related to placenta toxicity, describing the derangement of vascular placentation triggered by the activation of the AhR pathway and leading to pre-eclampsia, a pathology not only affecting mothers’ health but also proper fetus development. The other, AOP 431, describes the increased risk of gestational diabetes mellitus triggered by the induction of tumor necrosis factor. This in turn may also affect placenta functionality and thus fetus development.

### 4.10. Remarks

Among the AOPs herein described, only 28 were strictly related to developmental or mother/placental toxicity, the others being more general and whose applicability still has to be evaluated. Except for reproductive alterations, no consideration of the impact of sex/gender on modulating developmental toxicity responses has been paid. Since sex/gender-specific effects have been recently becoming more and more relevant for interpreting some of the variability observed in the response to toxicants, more efforts should be undertaken to develop sex-specific AOP and then specifying the sex affected in the Sex domain within AOP-Wiki.

## 5. Future Perspective of a Feto–Placental OoC System: Advantages and Potential

OoC platforms are novel technologies able to reproduce physiological functions of in vivo tissues more precisely than classical in vitro cell-based model systems [186]. They allow implementing dynamic co-cultures where the microfluidic flow, the sheer stress and other physical parameters are fully controlled, with the advantage to recreate more reliable organ cross talks.

Some placenta-on-chip devices were implemented co-culturing human trophoblasts (BeWo or JEG3) and endothelial cells (HUVECs or HPVECS) in order to mimic placental barrier physiology and transport [187,188,189]. Similarly, primary amnion epithelial and decidual cells were assembled in devices to recreate the feto–maternal interface [190]. A direct comparison of the feto–maternal interface OoC platform with other in vitro techniques (cell co-culture, transwell and placenta perfusion) has recently been discussed, highlighting the more physiological conditions provided by the OoC technology for barrier microfluidic studies [191]. None of these models included embryo/fetal cell types different from those directly facing the maternal side; however, some of them investigated adverse effects induced by exposure to nanoparticles [189], cadmium [192], cigarette smoke and dioxin [193].

To provide an efficient and informative developmental toxicity platform, the necessary step forward is the development of a “next generation” OoC integrating different human-relevant cellular models, not limited to the feto–maternal components of the placenta [194]. To reach this goal, a feto–placental-OoC (FPOoC) future system (Figure 2) should include: (i) a human placenta explant model from a pregnancy carrying a female or a male fetus and representing the elective barrier model to be subjected to chemicals’ exposure; (ii) downstream the placenta barrier, sex-matched female or male fetal amniotic membranes and fetal cell monolayers derived from different organs must be modularly placed to recreate an intrauterine environment. Moreover, by the culturing in a microfluidic flow, the fetal cells will come in contact with compounds and/or metabolites directly filtered by the placental barrier, thus more realistically mimicking a fetal exposure. Importantly, to limit sampling of maternal and fetal primary starting material, as well as to guarantee reproducibility and sufficient life span for chemical testing, both placental and fetal cells/membranes must be immortalized.

The importance of considering not only the embryonic/fetal component but also the placental and the amniotic ones for recapitulating the complete and correct natural developmental embryogenesis is highlighted by a recent in vitro “synthetic embryo” model proposed for the mouse obtained by co-culturing mESC and cells obtained from extraembryonic tissues [195]. These 3D self-organizing structures recapitulate in a more complete way the phases of morphogenesis till gastrulation to neurulation and early organogenesis. 

The fetal cells used to implement the FPOoC could be derived from discarded abortive material and should be established from organs relevant for the TG 414 and responsive to external and toxic stimuli, especially exposure to EDCs. Such cell types will also be relevant for their potential to uncover effects and pathways possibly linked with long-lasting chronic adult diseases, thus allowing not only the identification of EDCs and their MoA on the different organs in relation to sex but also the investigation of the mechanisms underlying the developmental origin of diseases that have such a global impact on human health.

The implementation of a platform that will accommodate more than a fetal cell type may further expand the potential of such FPOoC system for tissue interactions studies, especially when organs that physiologically form endocrine axes are included. Moreover, this kind of advanced model would allow to assess the effects of toxicants on the differentiation potential of cells by hitting at a precise sensitivity window during development corresponding to the pregnancy stage the cells have been derived from. Above all, the identification and characterization of the toxicological impact and MoA will be achieved in a sex-specific way.

The FPOcC system should be designed to be able to harvest both (1) the circulating medium, at desired points upstream and downstream of the barrier, and (2) the placental/fetal cells, to proceed with a battery of assays. A set of specific and targeted assays to address regulatory relevant readouts, possibly linked with the available developmental AOPs described above, as well as to novel ones to be developed, should be implemented. High throughput and high content screening must be performed and integrated with other available evidence to nurture new data on MoA not yet characterized. Further, a system with such design will be a valuable supporting tool for the assessment of mixture effects. In addition, the integration with in silico methods must be implemented to model fetal/placental physiology (pharmacokinetic/pharmacodynamic (PK/PD) modeling), stressors classification and in vitro to in vivo extrapolations. This may ultimately allow to define an IATA approach all in one place.

Overall, a future FPOoC system should bring several unique and innovative aspects at different levels, thus prompting a strong implementation in the field. In particular, such FPOoC platform should pursue the following objectives:To have a strong human relevance due to the choice of cellular models more closely recapitulating normal physiology and multicellular complexity.To be based on human cells representative of different fetal target organs, at the sensitive window of development, relevant for EDCs identification and characterization.To reconstruct a more physiological fetal environment, by adding placental and amniotic barriers to the system, thus mimicking metabolic, hormonal and immune crosstalk occurring during gestation.To enable a modular configuration of the placental downstream fetal organs to reproduce relevant in vivo axes.To be sex-specifically assembled, coupling placental cells from male or female pregnancies to amniotic and fetal cells of matched sex, for the identification of sex-specific effects of stressors, especially EDCs, and sex-related sensitivities to develop chronic diseases later in life.To represent an advanced, miniaturized NAM platform based on the integration of in vitro and in silico models, enabling high throughput and high content screening, as well as set up of fit-for-purpose readouts linked to regulatory relevant AOPs for the inclusion in an IATA approach.To strongly support the reduction of animal studies due to the human relevance and the transferability of the obtained results.

## 6. Conclusions

Concluding, a future FPOoC platform with such design features will have great potential to accelerate the obtainment of human-relevant results at molecular level, which is important to describe the regulatory required MoA, especially for EDCs, for which the adverse effect is a consequence, as introduced by the EC [196]. The reduction of experimental time and the acquisition of high-content, human-based data through a FPOoC system will shorten the decision-making process about specific substances, thus positively impacting the society by reducing the disease burden and national economies by cutting GDP-associated expenditure.

This in vitro–in silico coupled system, with a clear application in the regulatory toxicology, will also improve basic research in placental biology in the biomedical field where, as previously discussed, the actual human-based model has ethical concerns and lacks statistical power due to tissue procurement, making the placenta an under-researched human organ. This future platform will provide a powerful tool to perinatology to study molecular events behind placenta–fetal impairments and their effect in the adultness to trigger chronic diseases such as cardiovascular, inflammatory, metabolic and oncological disorders [7].

Such a highly defined and comprehensive model, with upgrading possibilities and setup by a multidisciplinary group of experts, will be able to impact the health research from toxicology toward a medicine helping to address the Developmental Origins of Health and Disease hypothesis.

## Figures and Tables

**Figure 1 ijerph-19-15828-f001:**
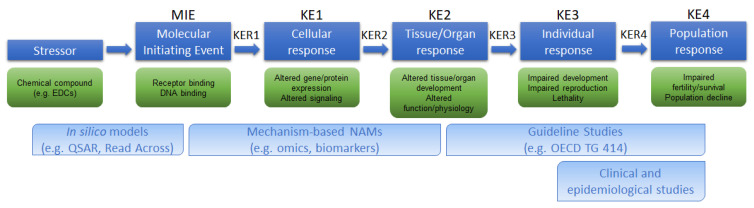
Schematic representation of the AOP conceptual framework. The Molecular Initiating Event (MIE) is triggered by a stressor and determines a cascade of key events (KE) causally linked by key events relationships (KERs). For each step, different investigation approaches may be applied according to the increasing complexity of the biological level observed.

**Figure 2 ijerph-19-15828-f002:**
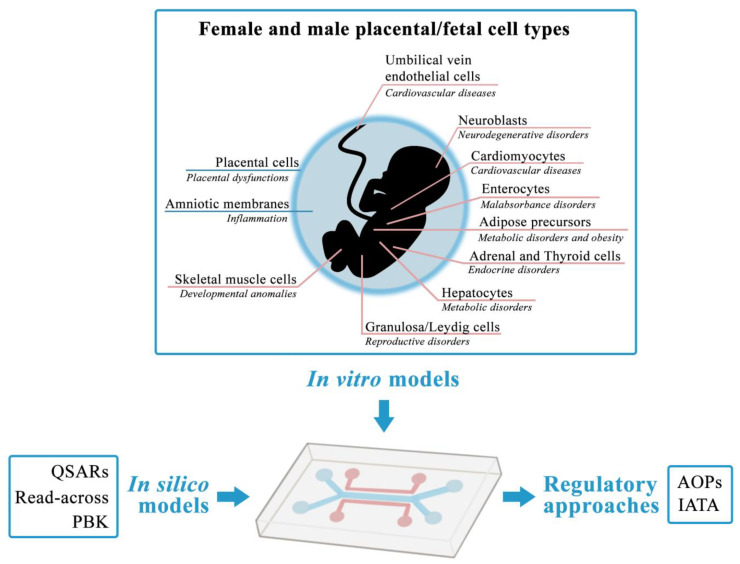
Requirements of a future feto–placental-OoC platform with the list of the cell type that can be implemented in the model. Fetal cells can be obtained from legal abortive material (9–12 weeks of gestation). A limited number of donors are required, also for amniotic and placenta cells. Immortalization of all these cell types will allow long-lasting availability of such cells. Possible associated chronic diseases which could be investigated through the model are indicated below each cell type.

**Table 1 ijerph-19-15828-t001:** Endpoints to be assessed in an OECD TG 414 test; the effects should be reported by the dose group.

Maternal Toxicity	Developmental Endpoints for Litters with Implants	Developmental Endpoints for Litters with Live Fetuses
Number of animals: at the start of the test, surviving, pregnant, aborting and delivering early	Number of *corpora lutea*	Number and percent of live offspring
Day of death during the study or whether animals survived to termination	Number of implantations	Sex ratio
Day of observation of each abnormal clinical sign and its subsequent course	Number and percent of live and dead fetuses and resorptions	Fetal body weight
Body weight, body weight change and gravid uterine weight, including body weight change corrected for gravid uterine weight	Number and percent of pre- and post-implantation losses	Anogenital distance of all rodent fetuses
Food and water consumption		External, soft tissue and skeletal malformations and other relevant alterations
For rat dams: thyroid hormones T4, T3 and thyroid-stimulating hormone		Total number and percent of fetuses and litters with any external, soft tissue or skeletal alteration
Thyroid histopathology assessment		
Necropsy findings, including uterine weight		
No Observed Adverse Effect Level (NOAEL)		

**Table 2 ijerph-19-15828-t002:** Examples of toxicological assays and outcomes on placental tissue explants.

Substance	Type of Placental Explant/Culture Period	Results	Ref.
BPA	Placental explant cultures	Increase of beta-hCG secretion and caspase-3 expression	[40]
Cadmium chloride	Term placenta explants, 24 h	Toxic effects	[41]
Cadmium chloride Mercury chloride	Term placenta explants, 24 h	Decrease in membrane fluidity and accumulation of cadmium in the membrane	[42]
DDT	Third-trimester human placental tissue explants, up to 72 h	Inhibition of CYP1A1 activity	[43]
Para-nonylphenol	First trimester chorionic villous explants, 24 h	Increase of cytokines at extremely low doses of p-NP	[44]

**Table 3 ijerph-19-15828-t003:** Pros and cons of the different alternative models for assessing developmental toxicity in humans.

	NAMs	PROs	CONs
1a	**Differentiated embryonic stem cell (ESC)**	Human models No ethical concerns for iPSCs Limited heterogeneity of sources Scaling up	Ethical concerns for ESC Single cell type not resembling the complexity of the organ Reduced differentiation potential as in vitro committed or reprogrammed Reduced sensitive window No systemic/surrounding environment effects & axis cross-talk No maternal and placental effect
1b	**Differentiated iPSCs**		
2	**Fetal/embryo primary cell organ cultures**	Human models Differentiation potential retained Intrinsic spatial organization Mixed populations resembling the organ of origin Reproducing the sensitive window	Ethical concerns Heterogeneity due to donors Limited scale up Limited time span Artificial exposure to chemicals No systemic/surrounding environment effects and axis crosstalk No maternal and placental effect

**Table 4 ijerph-19-15828-t004:** Examples of the use of hESCs, hiPSCs and fetal cells for developmental toxicological screening.

Toxicants	hESC-or ihPSC- Induced Lineage	Main Outcomes	Endpoints	Ref.
- Silver nanoparticles and AgNO_3_ 1b	hESC-derived cardiomyocytes and hepatocytes	Perturbed specification of two endoderm and mesoderm primary germ layers but not of ectoderm	Decreased expression of liver and cardiac markers	[92]
- Dioxin derivatives	hESC towards neural progenitor, early mesoderm and definitive endoderm cells	Lineage-specific regulation of Aryl Hydrocarbon Receptor (AhR) expression during human embryonic stem cell differentiation	Gene expression patterns of the lineage	[93]
- TCDD	hESC-derived cardiomyocytes	Inhibition of cardiomyocyte differentiation, mediated by AhR	Gene expression patterns of the lineage	[94]
- Cyclopamine - Valproic acid - Ochratoxin A - Mycophenolic acid - Theophylline and saccharin as negative control compounds	hESC-derived neural precursors	Alterations of neurogenic differentiation of hESC	Neural tube-like structures (rosettes) Expression of specific neuronal markers	[95]
- Penicillin-G - Caffeine - Hydroxyurea	hESC and embryoid bodies	Downregulation of markers associated with stemness, cardiac mesoderm, hepatic endoderm and neuroectoderm, indicating abnormal differentiation	Cell adherence Morphology Viability Apoptosis Lineage-specific expression markers	[96]
- 5-fluorouracil	iPSCs-derived neural precursors	Neurotoxicity via mitofusin-mediated mitochondria dynamics	Expression of neural differentiation marker genes Intracellular ATP content Mitochondrial fragmentation Mitochondrial fusion	[97]
- BPA - Tris(1,3-dichloro-2-propyl) phosphate - Cuprizone - Methyl mercury - Ibuprofen	BrainSpheres from hiPSC	Myelination process oligodendrocyte differentiation No toxicity	Gene expression analysis of myelination markers, cytotoxicity; myelin quantification	[85]
- 6-hydroxydopamine - 1-methyl-4-phenyl-1,2,3,6-tetrahydropyridine - 1-methyl-4-phenylpyridinium	BrainSpheres from hiPSC combined with endothelial barrier cells	Dopaminergic neuron toxicity	Gene expression analysis; metabolomics; ROS production; cytotoxicity	[86]
- Perfluorooctanesulfonic acid (PFOS) - Perfluorooctanoic acid (PFOA) - Ammonium 2,3,3,3-tetrafluoro-2-(heptafluoropropoxy)propanoate (GenX)	hiPSC line BiONi010-C-derived cardiomyocytes hiPSC line IMR90-1-derived cardiomyocytes	Reduced cardiac differentiation	PluriBeat assay to evaluate cardiac differentiation and cardiac gene expression markers	[98]
**Toxicants**	**Human Fetal Cells**	**Main Outcomes**	**Endpoints**	**Ref.**
- Benzo[a]pyrene	Human fetal olfactory epithelium neuroblasts	Impaired migration of gonadotropin-releasing hormone (GnRH) neurons	Gene expression of neuronal migration markers	[99]
- PCBs	Human fetal corpora cavernosa cells	Altered transcriptomic profiles	Toxicogenomics profiles Developmental pathways	[100]
- PBDEs	Human fetal progenitor cells from 16 weeks of gestation commercially available (NPCs)		Migration, differentiation, gene expression	[101]

## Data Availability

Not applicable.

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
