# Peer review of "Human-Based New Approach Methodologies in Developmental Toxicity Testing: A Step Ahead from the State of the Art with a Feto–Placental Organ-on-Chip Platform"

_ijerph, 2022, doi:10.3390/ijerph192315828_

Round 1

Reviewer 1 Report

This review entitled Human-based new approach methodologies in developmental toxicity testing: a step ahead from the state-of-the-art with a feto-placental organ-on-a-chip platform by Michaela et al is a comprehensive and detailed review article in which they highlighted the state of the art data on the availability of the in vitro (placental, fetal and amniotic cell-based systems) and in silico NAMs of human relevance for developmental toxicity testing purposes, as well as of the available Adverse Outcome Pathways related to developmental toxicity. The OECD TG 414 for the identification and assessment of deleterious effects of prenatal exposure to chemicals on developing organisms will be discussed to delineate the regulatory context and to better debate what is missing and needed in the context of the developmental origins of health and disease hypothesis to significantly improve this sector. Starting from this analysis, the development of a novel human feto-placental organ-on- chip platform will be introduced as an innovative alternative tool for developmental toxicity testing, considering possible implementation and validation strategies to overcome the limitation of the current animal studies and NAMs available in regulatory toxicology and in the biomedical field.

There are some topgraphical mistakes which should be addressed.

Abstract:

Line 31-32 should be changed its ambiguous.

Introduction:

Line 58-30. This sentence should be changed for the better understanding of the readers.

Line 104-116. Please rephrase this whole paragraph.

Overall the article has been well written and I would suggest these minor changes.

Author Response

Dear Reviewer,

please see the attachment for our point-by-point answers.

Thank you and regards

Reviewer 2 Report

The manuscript presented by Luconi et al., discusses the regulatory framework and the challenges and opportunities for developmental toxicology testing using human-relevant methodologies. This manuscript comprehensively covers the topics established in its objective, the review is written clearly, and it is organized in such a way that it is easy to read and follow the provided information. 

In Section 3, the Authors provide the reader with a thorough and critical review of current tools for studying developmental toxicology focused on humans. In addition, in sillico tools for studying developmental toxicology are discussed emphasizing strengths and weaknesses specific to each tool. The authors also propose a battery of adverse outcome pathways that can be associated with OECD TG 414 is proposed to strengthen the regulatory framework for developmental toxicology studies. Lastly, the manuscript emphasizes the significant advances that need to be made to better study and understand the feto-placental interface as a whole using an OoC platform. 

The abstract of the article suggests the idea of introducing a novel OoC feto-placental platform. However, in section 5, this new platform is presented as a relatively broad idea of what a complex evaluation system should and should not possess. Not enough concrete or conceptual ideas for the development of this OoC are presented. 

Also, to keep consistent their review structure, I consider valuable that the authors present a short introduction to the state of the art of OOC related to fetal, placental, and amniotic systems. 

Author Response

(The authors gave the same response as above.)
